# Linear Symmetric Quantization of Neural Networks for Low-precision Integer Hardware

**Xiandong Zhao[1,2], Ying Wang[1,3]\*, Xuyi Cai[1,2], Cheng Liu[1], Lei Zhang[1]**
Institute of Computing Technology, Chinese Academy of Sciences[1]
University of Chinese Academy of Sciences[2]
State Key Laboratory of Computer Architecture[3]
{zhaoxiandong,wangying2009,caixuyi18s,liucheng,zlei}@ict.ac.cn

## Abstract

With the proliferation of specialized neural network processors that operate on low-precision integers, the performance of Deep Neural Network inference becomes increasingly dependent on the result of quantization. Despite plenty of prior work on the quantization of weights or activations for neural networks, there is still a wide gap between the software quantizers and the low-precision accelerator implementation, which degrades either the efficiency of networks or that of the hardware for the lack of software and hardware coordination at design-phase. In this paper, we propose a learned linear symmetric quantizer for integer neural network processors, which not only quantizes neural parameters and activations to low-bit integer but also accelerates hardware inference by using batch normalization fusion and low-precision accumulators (e.g., 16-bit) and multipliers (e.g., 4-bit). We use a unified way to quantize weights and activations, and the results outperform many previous approaches for various networks such as AlexNet, ResNet, and lightweight models like MobileNet while keeping friendly to the accelerator architecture. Additional, we also apply the method to object detection models and witness high performance and accuracy in YOLO-v2. Finally, we deploy the quantized models on our specialized integer-arithmetic-only DNN accelerator to show the effectiveness of the proposed quantizer. We show that even with linear symmetric quantization, the results can be better than asymmetric or non-linear methods in 4-bit networks. In evaluation, the proposed quantizer induces less than 0.4% accuracy drop in ResNet18, ResNet34, and AlexNet when quantizing the whole network as required by the integer processors.

## 1 Introduction

Deep neural networks have shown excellent performance on various computer vision and natural language processing tasks, such as classification (Krizhevsky et al., 2012; Simonyan & Zisserman, 2015; He et al., 2016), object detection (Girshick, 2015; Redmon et al., 2016; He et al., 2017), segmentation (Long et al., 2015; Noh et al., 2015), machine translation (Zhang et al., 2018b), speech recognition (Nassif et al., 2019), etc. While the past few years witnessed the success of DNNs on cloud and server-end computers, neural networks have been recently pushed to embedded and mobile areas to enable edge intelligence. For these scenarios, the power provision and computational strength on the edge computing devices are limited. As a result, it is essential to have more efficient network architectures and less expensive inference overhead. Therefore, there is increasing attention from the research community to study the compression of modern deep neural networks that are typically over-parameterized and computationally costly.

Several categories of approaches are proposed to decrease the computational overhead of neural networks, such as lightweight neural network architectures (Howard et al., 2017), neural architecture search (NAS) (Elsken et al., 2018), and network pruning (Han et al., 2015; 2016; Wen et al., 2016;

---

*Corresponding author

Table 1: Comparison between different quantizers: **All-Layer (AL)** denotes quantizing all the parameters of all the operators in networks, including weights, bias, activations, and the scaling factor for low-precision networks; **BN** donates that the BN operation is only invoked in training but merged into weights and induces no overhead in integer inference; **Linear-Symmetric (LS)** denotes linear symmetric quantization; **Activation Functions (AF)** donates the support of *Leaky ReLU* and activation functions besides *ReLU*. **Structure-Intact (SI)** indicates the network structure is unmodified.

| Method | AL | BN | LS | AF[e] | SI |
|---|:---:|:---:|:---:|:---:|:---:|
| Deep Compression (Han et al., 2016)[b] | | | | | ✓ |
| WQ (Park et al., 2017)[b] | | | | | ✓ |
| LQ-Nets (Zhang et al., 2018a) | | | | | ✓ |
| Min-Max Linear Quantization[a] | | | | ✓ | ✓ |
| DoReFa (Zhou et al., 2016)[c] | | | | ✓ | ✓ |
| RQ (Louizos et al., 2019) | | | ✓ | ✓ | ✓ |
| WRPN (Mishra et al., 2018) | | ✓ | ✓ | ✓ | |
| PACT (Choi, 2018)[d] | | | ✓ | ✓ | ✓ |
| LLSQ(ours) | ✓ | ✓ | ✓ | ✓ | ✓ |

[a] Naive linear quantization, which finds min-max value at runtime.
[b] Clustering-based approaches to quantize weights.
[c] DoReFa falls into linear asymmetric quantizer due to the need for offset.
[d] In Choi (2018), they use PACT to quantize activations, and DoReFa to quantize weights.
[e] DoReFa, RQ, WRPN, and PACT are designed for *ReLU*, but they can be extended to support other activation functions in theory.

Molchanov et al., 2017). Besides these techniques, quantizing high-precision floating-point networks to lower bitwidth representation can also drastically decrease both the static parameters and the intermediate data generated during the network inference, resulting in reduced memory footprint and also computational intensity. And this paper focuses on the quantization of neural networks.

Quantization technique is also closely related to the implementation of specialized hardware that maps the procedure of network inference onto the energy-efficient low-precision integer or fixed-point arithmetic circuits. In the hardware perspective, low-precision integer accelerators or processors are dominating the solutions targeted on neural network inference, especially for mobile and embedded scenarios. Google's Tensor Processing Unit 1.0 (TPU) (Jouppi et al., 2017), Unified Deep Neural Network Accelerator (UNPU) (Lee et al., 2018), Eyeriss (Chen et al., 2018), Stripes (Judd et al., 2016), Pragmatic(Albericio et al., 2017) and many other newly proposed hardware implementations are generally reliant on the effectiveness of the underlying quantization techniques, which are especially crucial for the low-precision integer hardware designed to process binary, ternary, 4-bit or 8-bit networks. In other words, quantization is not only a method to reduce the memory footprint as in traditional work, but also a mandatory step to make the network deployable on integer hardware.

Though there is a lot of prior work that investigates low-precision quantization, they mainly target on reducing the memory overhead caused by floating or high precision data representation in the networks, but not focus on specialized integer hardware for network inference. To enable the neural network processors to work with low-precision integer operands and minimize the accuracy losses, a good network quantizer must satisfy the constraints as enlisted in Table 1.

**First**, all the parameters, including weights, bias, activations, partial results that eventually accumulate to an activation, and even the scaling factors, which are indispensable for low-precision networks like binary and ternary representation, must be quantized into low bitwidth integers as required by the underlying specialized hardware. In some prior work (Zhou et al., 2016; Zhu et al., 2017; Zhang et al., 2018a; Mishra et al., 2018; Choi, 2018), they either leave bias and scaling factors unquantized or keep the first and last layer in full or high precision. Besides, some designs rely on high-precision internal register or ALUs to support high-precision partial results that are generated during computation before the final output of activations or features. For example, Krishnamoorthi (2018), which quantizes the weights and activations to 8-bit, directly use 32-bit accumulators to cache the intermediate values or partial results to avoid overflows. However, for 4-bit and lower bitwidth, the integer accelerators cannot afford high bitwidth accumulators, which indicates higher silicon area and power cost. For integer-only-arithmetic, we quantize the bias to fixed-point numbers by using a straight-forward method. The value range of these numbers is wide, resulting in overflows of the

low bitwidth accumulators. To overcome this problem, we quantize the bias to 8-bit and finetune the bias of the model. As shown in Figure 1, the bitwidth of accumulators can be reduced to 16-bit.

**Second**, the *BatchNorm (BN)* layer does not necessarily need to be processed during inference for the reduction of computation and memory cost. For most of the convolutional neural networks, *BN* layers are often after the *Conv* or *FC* layers. In these situations, *BN* can be merged into the weights and biases of the corresponding *Conv* or *FC* layers. However, in Zhou et al. (2016); Zhang et al. (2018a), they use asymmetric or non-linear quantization, causing barriers to *BN* fusion. There are two ways to overcome this obstacle. One is "*BN* folded training"(Krishnamoorthi, 2018), which adopts *BN* fusion before weights quantization in every training step; the other is to use symmetric linear quantization. However, the first method doubles the training time, while the second one has no additional computational overhead, which will be introduced in Section 3.4.

**Third**, linear quantization is necessary for state-of-the-art accelerators. There are many non-linear quantization methods which achieve excellent bitwidth reduction efficacy and accuracy tradeoffs. In these cases, it requires additional transformation to have correct arithmetic results after quantizing the value into non-linear distribution. For example, as in Han et al. (2016); Park et al. (2017), it necessitates the operation of table lookup to have correct multiplication between quantized values. However, the linear quantization can make full use of the low-precision arithmetic components in off-the-shelf accelerators. Further, linear quantization can be divided into symmetric mode and asymmetric mode. Asymmetric quantization has one more parameter (e.g., zero-point (Krishnamoorthi, 2018)) than symmetric quantization, and it requires additional subtraction or linear-operation before multiplication. As a result, the symmetrical mode is compatible with the mainstream integer accelerator chip design and do not require the redesign of datapath in these hardware.

**Fourth**, different CNNs or applications usually use a variety of activation functions. For instance, the object detection model Redmon et al. (2016) typically uses *Leaky ReLU*. And the bottleneck of ResNet block does not use any activation function. The quantization methods are expected to be adapted to these situations. However, Zhang et al. (2018a); Park et al. (2017) only focus on the quantization of activations after *ReLU*. In this paper, we demonstrate our method is friendly to different activation methods such as *Leaky ReLU*.

Some of the previous researches change the network structure for better quantization performance, e.g., Mishra et al. (2018) double or even triple the convolutional filters to reduce accuracy degradation. For the energy-efficient integer neural network chips, it needs to remap the changed network architecture to hardware and adds to computational and memory access overhead due to the increased filters and parameters. As a result, keeping the network structure intact is important.

Concerning all the factors above, in this paper, we present a learned linear symmetric quantization (LLSQ) method and also evaluate it on a low-precision neural network accelerator through hardware-software co-design. Specifically, our mainly contributions are:

- Unlike most of other quantization methods, we quantize the whole network including the first and last layers. We also quantize bias and scaling factors, in support of the low bitwidth integer arithmetic units and accumulators on the accelerator.

- We adopt learned linear symmetric quantization schemes which are hardware friendly (such as the convenience of *BN* fusion implementation) while achieving state-of-the-art prediction accuracy.

- We design a specialized low-precision CNN inference accelerator to validate the methodology, which supports 2/4/8 integer operating and work with high efficiency. We then deploy our quantization model on the accelerator to illustrate the efficacy of the workflow.

## 2 MOTIVATION

Edge or embedded neural network accelerators generally have three primary design goals— small-footprint, high-throughput/low-latency, and low-power. For different applications and scenarios, the prior researches on specialized deep learning processors are often falling into different categories: cloud-oriented hardware for warehouse machines, low power mobile processors and ultra-low power accelerators for IoT or cyber-physical devices.

For mobile and embedded usage, specialized neural network processors are becoming increasingly popular as an efficient hardware solution of inference. DianNao (Chen et al., 2014) is proposed for fast inference of DNNs and it uses 16-bit fixed-point multipliers for small silicon area and low-energy. Later, ShiDianNao (Du et al., 2015) is introduced and it burns extremely low energy consumption by putting all weights onto the SRAM to eliminate considerable DRAM accesses. Besides, DeepBurning (Wang et al., 2016) simplifies the design flow of accelerator for different NN models. Eyeriss (Chen et al., 2018) is also another representative of low-power accelerators. And it presents a row-stationary (RS) dataflow to minimize data movement energy consumption on a spatial architecture. To further reduce computation overhead, EIE (Han et al., 2016) exploits the sparsity and low-bit compression of the NNs and achieves better throughput, energy and area efficiency. These typical edge neural network processors are accepting fixed-point data input and using fixed-point processing elements to reduce the power and chip area overhead caused by floating-point arithmetic components and memory. For the cloud scenarios, specialized architectures like TPU (Jouppi et al., 2017) and FPGA-based accelerator cards are also replacing conventional GPGPU and CPU for high-throughput inference tasks. Even for cloud-oriented inference architectures, fixed-point processing architectures like TPU are favored because they are able to deliver much higher throughput for the given power budget and silicon area overhead.

However, for the fixed-point or integer hardware targeted on neural network acceleration, quantization is prerequisite to convert the floating-point network model into the fixed-point format compatible with the specialized hardware, and it is also a critical step to ensure the accuracy of the network after conversion. Many prior quantization methods are intended to reduce the running overhead of networks but ignore the architecture and working mechanism of integer neural network processors, as illustrated in Table 1, and they sometimes face considerable accuracy losses, or performance penalty or even fail to be supported on the realistic integer datapath due to the unconsciousness of the underlying hardware. This problem becomes particularly important for the hardware that is designed to run low bitwidth networks such as binary, ternary, and 2/4-bit models. For instance, Deep compression and WQ are clustering-based quantization methods, and they still need high-precision values to represent the weights, bias, and activations. As a result, they are not compatible with the hardware that only supports low-precision computing. LQ-Nets uses non-linear quantization based on the binary code and basis vector, and it can theoretically calculate the inner products between quantized weights and activations by bitwise operations only. However, it requires intensive modifications to the design of current processors by adding a lot of look-up tables in the datapath. Further, bias and scaling factors are not quantized in PACT and WRPN, resulting in performance penalty when employing additional high-precision or float-point ALUs to deal with them. In contrast, our LLSQ is designed to ease the model quantization flow for the specialized integer neural network processors by conforming to the constraints specified in Table 1. To validate the importance of hardware-aware quantizer and software/hardware co-design, we also design a specialized CNN accelerator for wearable applications. And the specialized accelerator supports 2/4/8 integer operation and adopts the dataflow of low latency and energy design.

## 3 Networks with Learned Linear Symmetric Quantization

In this section, we firstly give the overview of the proposed quantization scheme. Then we detail the scheme including low-precision representation, quantized network training, and the deployment of quantization model on our specialized integer-only CNN accelerator for fast inference.

### 3.1 Overview of LLSQ

Many of the previous researches focus on the quantization-aware training in GPU, showing the potential of low-bit quantization on CNNs. Han et al. (2016); Park et al. (2017); Zhang et al. (2018a) propose non-linear quantization methods but lacks of a detailed description of the hardware feasibility. Krishnamoorthi (2018) provides a quantization scheme that quantizes weights and activations into 8-bit integers and integer-arithmetic-only implementation on ARM CPUs. The method achieves evident hardware acceleration effects, but does not fully exploit lower-precision quantization. Based on the researches, we propose a quantization scheme for state-of-the-art specialized accelerators operating on low-precision integers only. Figure 1 shows an overview of the proposed scheme.

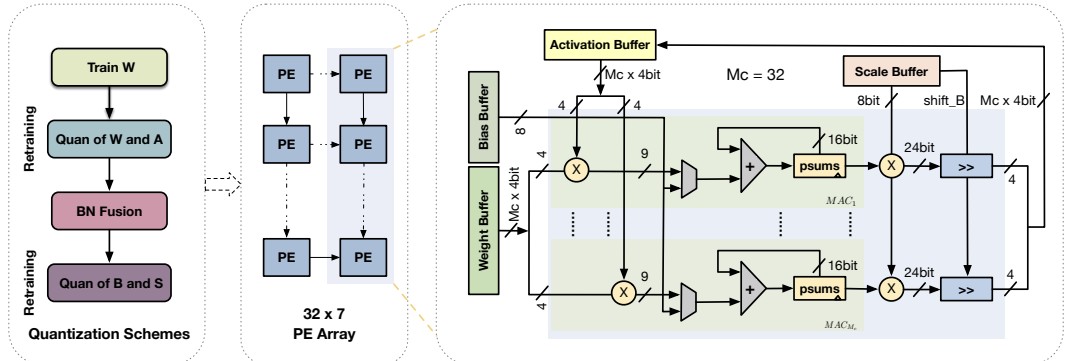

Figure 1: An overview of LLSQ: using pre-trained weights for fast convergence; Retraining of the network with quantized weights and activations; *BN* fusion for efficient inference; Quantization of bias and scaling factors; Deployment of the quantized model to our accelerator. As shown in this figure, weights, activations, bias, and scaling factors are quantized to low-bit integers. And the bandwidth of accumulator can be set to lower (e.g., 16-bit in our experiments).

Compared with prior work, our proposed quantization scheme pays more attention to the constraints imposed by real hardware. We use a unified learned linear symmetric quantizer to quantize weights and activations. And the quantizer has only one parameter, known as the scaling factor. Linear symmetric quantization consumes little additional resources based on the mainstream integer accelerator designs while achieving state-of-the-art accuracy in various networks. After that, we adopt *BN* fusion for fast inference on hardware. As for bias and scaling factors, we also quantize them to low-bitwidth integers. The integer accelerator illustrated in Figure 1 is an illustrative case of 4-bit quantization and hardware acceleration.

## 3.2 MAKING FULL USE OF THE PRE-TRAINED PARAMETERS

In experiments, we find that it is more efficient to start with the pre-trained full-precision parameters before quantization. Louizos et al. (2019); Zhou et al. (2017) use pre-trained weights for fast convergence and deployment, while Zhang et al. (2018a); Choi (2018); Cai et al. (2017) train quantized network from scratch to show the robustness of the algorithm. However, for some object detection models, the backbone models and pre-trained weights are essential to the detection performance. Redmon et al. (2016) shows that the pre-trained high-resolution classification network gives an increase of almost 4% *mAP*. To have better performance in classification, object detection, and other CNN based tasks, in this paper, we use pre-trained parameters to initialize the networks.

## 3.3 LOW-PRECISION REPRESENTATION AND QUANTIZATION ALGORITHM

We use channel-wise quantization for *Conv* layers and layer-wise quantization for *FC* layers and activations. And we adopt the symmetric linear quantization to quantize weights or activations into $k$ bits words(e.g., 4-bit), which can be defined as

$$
\begin{aligned}
\boldsymbol{x}^q &= Quantize_k(\boldsymbol{x}^r \mid \alpha) \\
\boldsymbol{q} &= \frac{\boldsymbol{x}^q}{\alpha} = clamp(\lfloor \frac{\boldsymbol{x}^r}{\alpha} \rceil, -2^{k-1}, 2^{k-1} - 1)
\end{aligned}
\tag{1}
$$

where $\boldsymbol{x}^r \in \mathbb{R}$ is one kernel of weights or one layer of activations, the variable $\alpha \in \mathbb{R}^+$ is the quantization parameter, known as the scaling factor, while $\boldsymbol{q} \in \{-2^{k-1}, \ldots, 0, 1, \ldots, 2^{k-1} - 1\}$ is the integer values flowing in the integer accelerator and $\boldsymbol{x}^q \in \{-2^{k-1}\alpha, \ldots, 0, \alpha, \ldots, (2^{k-1}-1)\alpha\}$ is the quantized weights or activations. Note that for activations, which are non-negative values if the ActFun is *ReLU*, we clamp them to $[0, 2^k - 1]$, resulting in $\boldsymbol{q} \in \{0, 1, \ldots, 2^k - 1\}$ and $\boldsymbol{x}^q \in \{0, \alpha, \ldots, (2^{k-1} - 1)\alpha\}$, respectively.

As defined above, we use $\alpha$ as our quantization parameter. And we optimize it with:

$$
\alpha^* = \arg\min_{\alpha} \int p(\boldsymbol{x}^r)|\boldsymbol{x}^q - \boldsymbol{x}^r|^l
\tag{2}
$$

Table 2: Comparison of SG and EMA.

| Method | VGGSmallw4a4 | VGGSmallw2a2 | ResNet18w4a4 | ResNet18w3a3 |
|--------|--------------|--------------|--------------|--------------|
| EMA | 93.95 | 92.78 | 69.48 | 66.80 |
| SG | **94.34** | **93.31** | **69.84** | **68.08** |

VGGSmall is trained on Cifar10 and ResNet18 is on ImageNet.

where $\boldsymbol{x}^r$, $\boldsymbol{x}^q$ are the same factors defined in Equation 1, $p(\boldsymbol{x}^r)$ is the probability density distribution of $\boldsymbol{x}^r$, and $l \in \{1, 2\}$ is an optional constraint (We use 2 in our experiments). In Figure 2, we present the relationship between quantization error and $\alpha$. When fixing weights $\boldsymbol{x}^r$, we can find the optimal $\alpha^*$ by using the brute-force search approach, which induces high computation cost. Besides, the weights are updated during the re-training phase and the optimal value $\alpha^*$ changes accordingly. In other words, the optimal value for the factors is not fixed and it takes considerable computational overhead to find the dynamic optimal value.

Inspired by Zhang et al. (2018a), we find through experiments that there is no need to find the optimal value $\alpha^*$, and it works well enough to find a near-optimal value $\tilde{\alpha}^*$. Generally, quantization can be considered as a regularization of the networks, and the quantization parameter $\alpha$ needs only to be adjusted to a near-optimal value to preserve the network capacity.

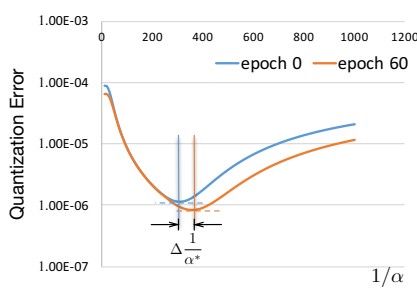

Then the problem becomes how to find a $\tilde{\alpha}^*$ in the forward pass of the network training phase. At the beginning of training, we assign $\alpha$ an initial value. Then in every training iteration, we explore between $2\alpha$ and $\alpha/2$ to find a better search direction $d_{better} \in \{-1, 0, 1\}$, and use $-\alpha^2 d_{better}$ as the simulated gradient (SG) of $\alpha$ which is detailed in Equation 9. The gradients of other parameters are still obtained by backpropagation algorithm. After that, we update all parameters with the gradients or simulated gradients. Another method is updating $\alpha$ by the exponential moving average (EMA). We experiment both of the methods, and the results show that SG is generally better than EMA on various networks (See Table 2). If not specifically stated, we use the SG method in experiments. The re-training process with weights and activations quantized is summarized in Step 1 of Algorithm 1.

Figure 2: L2 distance of quantization. The data is from weights of the first FC layer in AlexNet. As shown in the figure, the optimal $\alpha^*$ changes with the updating of weights.

### 3.4 BN Layer Fusion of Quantized Networks

As described in Section 1, merging the *BN* layers into convolutional layers can reduce the latency of network inference by removing additional computation overhead. The operator of quantized *Conv*[1] and *FC* layers can be expressed as

$$\boldsymbol{o} = \alpha_a \boldsymbol{q}_a \alpha_w \boldsymbol{q}_w + b \tag{3}$$

where $\alpha$, $\boldsymbol{q}$ are the same as Equation 1, $\alpha_a \boldsymbol{q}_a$, $\alpha_w \boldsymbol{q}_w$ donate the quantized activations and weights, while $b$ is the bias and $\boldsymbol{o}$ is the output feature vector. Note that $\alpha_a$, $\alpha_w$ and $b$ are full precision values. And the *BN* layer can be formulated as follows:

$$\boldsymbol{y} = \frac{\boldsymbol{o} - \mu}{\sqrt{\sigma^2 + \epsilon}} \gamma + \beta \tag{4}$$

where $\mu$ and $\sigma^2$ are EMA statistics, $\gamma$ and $\beta$ are learned parameters in *BN* layers.

Obviously, we can merge *BN* layers and figure out the corrected parameters:

$$\hat{\alpha}_w = \zeta \alpha_w; \quad \hat{b} = (b - \mu)\zeta + \beta; \quad \zeta = \frac{\gamma}{\sqrt{\sigma^2 + \epsilon}} \tag{5}$$

---

[1]For brevity, we only consider the operation of one channel.

### 3.5 BIAS AND SCALING FACTOR QUANTIZATION FOR LOW-BIT INTEGER ONLY ARITHMETIC

Further, the outputs of layers are quantized according to Equation 1. For integer-only-arithmetic, the bias use $\alpha_a \alpha_w$ as its scaling factor. And for the multiplier $\frac{\alpha_a \alpha_w}{\alpha_o}$, we use bit-shift quantization (See Equation 10) so that no multiplication but bit-shift operation is needed in hardware.

$$\alpha_o \boldsymbol{q}_o = \alpha_a \boldsymbol{q}_a \hat{\alpha}_w \boldsymbol{q}_w + \hat{b}$$

$$\boldsymbol{q}_o = \frac{\alpha_a \hat{\alpha}_w}{\alpha_o} (\boldsymbol{q}_a \boldsymbol{q}_w + q_b) \tag{6}$$

$$where\ q_b = clamp(\lfloor \frac{\hat{b}}{\alpha_a \hat{\alpha}_w} \rceil, -2^{k_b - 1}, 2^{k_b - 1} - 1)$$

Note that $\alpha_a \alpha_w$ is a very small number, resulting in large quantization noise when adopting the clamp operation. In addition, the quantization of the scaling factors $\alpha$ can also raise the quantization noise of weights and activations. Parameter re-training summarized in Step 2 of Algorithm 1 is required.

In the re-training phase, we adopt STE (Bengio et al., 2013) to realize the non-differentiable quantization function.

For weights and bias, we have

$$\frac{\partial y}{\partial w^q} \simeq \frac{\partial y}{\partial w^r};\ \frac{\partial y}{\partial b^q} \simeq \frac{\partial y}{\partial b^r} \tag{7}$$

For activations, we have

$$\frac{\partial y}{\partial a^q} \simeq \begin{cases} \frac{\partial y}{\partial a^r} & if\ 0 \leq a^r \leq (2^k - 1)\alpha \\ 0 & otherwise \end{cases} \tag{8}$$

## 4 EXPERIMENTAL RESULTS

In this section, three sets of experiments on Cifar10, ImageNet and Pascal VOC datasets are presented. First, we conduct our proposed learned linear symmetric quantization (LLSQ) on weights and activations, leaving the first and last layers in full precision for a fair comparison with Zhang et al. (2018a). Second, we quantize the whole networks including the first and last layers, which is referred as LLSQF (LLSQ for Full network). Finally, we quantize the remaining bias and scaling factors. LLSQ is implemented in PyTorch (Paszke et al., 2017), and most of the baselines it uses in evaluation are from PyTorch Model Zoo[2].

### 4.1 QUANTIZATION OF WEIGHTS AND ACTIVATIONS

We firstly employ the VGG-Small network on Cifar10 to verify the LLSQ method. After that, we use AlexNet (Krizhevsky et al., 2012), ResNet18, ResNet34 (He et al., 2016), particularly the light-weight and hard-to-compress network architecture of MobileNet (Howard et al., 2017; Sandler et al., 2018) etc. to conduct more detailed experiments on the ImageNet dataset. Finally, we also quantize YOLOv2 (Redmon & Farhadi, 2017) to demonstrate that LLSQ also works well for complicated applications and especially the task adopting the activation functions like *Leaky ReLU* other than *ReLU* used in previous work.

VGG-SMALL ON CIFAR10

The VGG-Small architecture is the same with Louizos et al. (2019); Zhang et al. (2018a), consisting of six *Conv* layers, three *MaxPool* layers, and one *FC* layer. We adopt a cosine learn rate scheduler to train the VGG-Small reference and the quantized models. Specifically, we train the reference network for 400 epochs using an initial learning rate of 2e-2. And for the training of the quantized network, we use a warmup learning rate scheduler in the first ten epochs with an initial learning rate of 2e-3. In all quantization experiments, the total training epochs are 100. The VGG-Small quantization results are provided in Table 3. With 3-bit weights and 3-bit activations, the accuracy

---

[2]https://pytorch.org/docs/stable/torchvision/models.html

Table 3: Comparison with the state-of-the-art low-bit quantization methods on CIFAR-10. The bitwidth for weights($w$), activations($a$), bias($b$) and scaling factor($\alpha$) are given.

| Method | # Bits $w/a/b/\alpha$ | Acc(%) | Degradation(%) |
|---|---|---|---|
| LQ-Nets[*] | Reference | 93.8 | |
| (Zhang et al., 2018a) | 3/3 | 93.8 | 0.0 |
| | 2/2 | 93.5 | 0.3 |
| RQ | Reference | 93.05 | |
| (Louizos et al., 2019) | 8/8 | 93.30 | -0.25 |
| | 4/4 | 91.57 | 1.48 |
| | 2/2 | 90.92 | 2.31 |
| LLSQ[*](ours) | Reference | 93.34 | |
| | 4/4 | **94.34** | -1.00 |
| | 3/3 | **94.02** | -0.68 |
| | 2/2 | 93.31 | 0.03 |
| LLSQF(ours) | 4/4 | 94.30 | -0.96 |
| | 3/3 | 94.07 | -0.73 |
| | 2/2 | 93.12 | 0.22 |
| | 4/4/8/8 | 93.84 | -0.50 |

[*] first and last layer in full precision

using our method is better than state-of-the-art method, LQ-Nets. And even when the first and last layers are all quantized in the same way, the loss of accuracy is minimal.

### IMAGENET DATASET

We then quantize AlexNet, ResNet18, ResNet34 and MobileNetv2 on ILSVRC2012 dataset with different bitwidth configuration to demonstrate the effectiveness of the method. All of the pre-trained float-point weights except MobileNetv2[3] are downloaded from the PyTorch Model Zoo, and they are trained for 90 epochs with a step learning rate scheduler. After loading the pre-trained weights, we employ a warmup learning scheduler in the first three epochs and the cosine scheduler in the remained 57 epochs with an initial learning rate of 2e-2.

As shown in Figure 3a, when quantizing both weights and activations, our degradation of accuracy is significantly smaller than LQ-Nets, PACT, and RQ. Especially, LLSQ outperforms the baselines when quantizing weights and activations into 4-bit. And it also outperforms other non-linear quantization methods with different bitwidth. Figure 3b shows that even with the first and last layers quantized, it can still achieve near baseline performance. In overall, the accuracy drop is less than 0.4% in ResNet18, ResNet34, and AlexNet when quantizing the whole network. We also quantize MobileNetv2, a more compact network, and obtain results that are significantly better than RQ. Please check Table 7 for detailed results.

### OBJECT DETECTION ON PASCAL VOC

We also apply the proposed LLSQ to YOLOv2. The backbone of YOLOv2 is Darknet-19, and its activation function is *Leaky ReLU*, so that the activations contain negative values. For YOLOv2 on Pascal VOC, we adopt the same quantization configuration (See Section 3.3) of the weights to the activations. Results are listed in Table 4. As shown in the table, LLSQ induces minor losses of *mAP* in different bitwidth presentation. For comparison, we also quantize the activations into signed 5-bit integers using PACT, and consequently face considerable mAP losses (54.8*mAP*). Please note that we use the open-source PyTorch implementation of YOLOv2 [4] as the baseline. We train the quantized model for 170 epochs (2/3 of baseline) with an initial learning rate of 1e-4 (1/10 of baseline).

---

[3]https://github.com/tonylins/pytorch-mobilenet-v2
[4]https://github.com/marvis/pytorch-yolo2

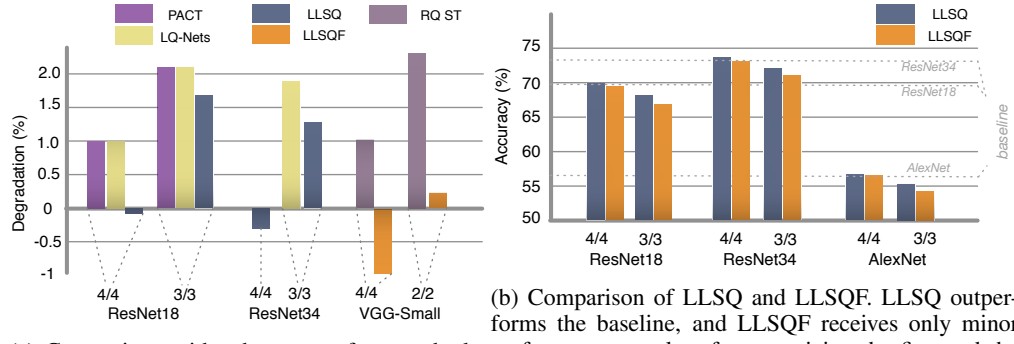

(a) Comparison with other state-of-art methods. Lower is better.

(b) Comparison of LLSQ and LLSQF. LLSQ outperforms the baseline, and LLSQF receives only minor performance penalty after quantizing the first and the last layers.

Figure 3: Quantization results on different networks.

Table 4: LLSQ on YOLOv2 detector.

| bitwidth | mAP | aero | bike | bird | boat | bottle | bus | car | cat | chair | cow | table | dog | horse | mbike | person | plant | sheep | sofa | train | tv |
|---|---|---|---|---|---|---|---|---|---|---|---|---|---|---|---|---|---|---|---|---|---|
| FP32 | 73.2 | 78.9 | 80.0 | 72.8 | 62.3 | 47.1 | 79.2 | 79.6 | 85.7 | 54.4 | 79.7 | 72.2 | 83.3 | 81.1 | 79.2 | 74.8 | 48.4 | 75.7 | 72.3 | 83.4 | 73.0 |
| w4a5 | 70.3 | 73.9 | 76.1 | 67.8 | 57.3 | 39.9 | 81.2 | 79.1 | 82.6 | 51.8 | 75.7 | 68.3 | 80.3 | 83.9 | 78.7 | 70.6 | 42.6 | 72.2 | 71.6 | 83.5 | 69.5 |
| w32a5 | 71.2 | 75.5 | 75.9 | 71.4 | 60.4 | 42.4 | 80.6 | 80.0 | 83.3 | 53.5 | 75.8 | 68.1 | 70.8 | 82.6 | 79.5 | 71.6 | 45.5 | 69.9 | 72.1 | 84.6 | 70.4 |
| w4a8 | **73.4** | 74.5 | 79.1 | 75.5 | 60.6 | 43.8 | 80.9 | 80.7 | 85.8 | 56.6 | 80.0 | 70.9 | 83.5 | 84.5 | 81.0 | 74.5 | 47.5 | 74.8 | 75.2 | 84.1 | 73.6 |
| w4a32 | **74.2** | 74.6 | 78.6 | 75.5 | 66.0 | 47.4 | 80.8 | 83.2 | 87.4 | 57.3 | 80.3 | 70.8 | 83.7 | 84.3 | 83.0 | 74.8 | 49.4 | 74.2 | 73.8 | 85.2 | 73.5 |

## 4.2 *BN* FUSION AND QUANTIZATION OF BIAS AND THE SCALING FACTOR

We adopt *BN* fusion in the PostAct (*Conv→BN→ReLU*) networks according to the formula in Section 3.4. And the scaling factor of bias is the product of the corresponding scaling factors belonging to the activations and the weights, respectively. After that, we visualize the bias value distribution of VGG-Small. Figure 4 shows $b/\alpha_b$ is distributed between a vast range (-1000, 1000), resulting in overflows of low bitwidth accumulators. And the overflow phenomena have a significantly harmful impact on the network performance. To deal with this issue, we quantize the bias and the scaling factors to 8-bit words, and then fine-tune the networks to restore the original performance. Generally, we need fine-tuning for one epoch only. After the quantization of bias and scaling factor, we have a fully quantized model and have it deployed onto our integer-only accelerator with 16 bitwidth accumulators. Table 3 and 7 show that the accuracy loss is negligible with $w4a4b8\alpha8$ quantization on both VGG-Small and AlexNet.

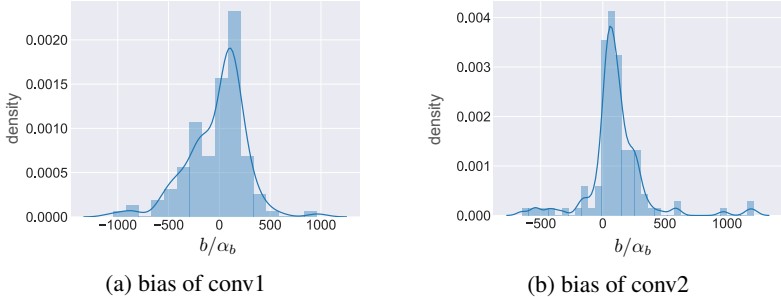

(a) bias of conv1        (b) bias of conv2

Figure 4: Distribution of the bias/scaling factor. The data is from VGG-Small with w4a4 quantization.

## 4.3 DEPLOYMENT ONTO REALISTIC HARDWARE

The introduced linear symmetric quantization is intended to deploy the quantized networks to specialized integer-only-arithmetic CNN accelerators or other integer-only hardware. Our accelerators adopt the typical 2D systolic array architecture (Chen et al., 2018), but they are featured with 4-bit or 2-bit low-precision operation. As shown in Figure 1, the 8/4/2-bit accelerator has a 32x7 array of

Table 5: Comparison of our low-precision integer Neural Network Processors.

| Bitwidth | #MAC Unit | Throughput (GOps/sec) | Silicon Area (mm$^2$) | Power (mW) |
|---|---|---|---|---|
| 8-bit | 224 | 179.2 | 4.71 | 228 |
| 4-bit | 224 | 179.2 | **2.80** | **93** |
| 2-bit | 224 | 179.2 | **1.84** | **41** |

Implemented and synthesized with Synopsys Design Compiler (DC) under the 40nm technology.

processing elements (PE). And the MAC unit in each PE consists of a 4-bit multiplier and a 16-bit accumulator. For the 4-bit accelerator, we use INT4 representation for weights, UINT4 for activations, INT8 for the bias and scaling factors, respectively. For the 2-bit accelerator, we use INT2 for weights, UINT2 for activations, INT8 for the bias and scaling factors, respectively. Through the quantization process described in the paper, we can have a fully quantized network that works directly on the CNN accelerator. In addition, as we use linear symmetric quantization, we can use a straight-forward way to conduct multiply-accumulate operations without introducing shifters or lookup tables, which means the quantized models can run on state-of-the-art integer accelerators and ensures that their output accuracy degradation is minimal as presented in the above sections. Finally, we implement the 8/4/2-bit integer neural network processors with Synopsys Design Compiler (DC) under the 40nm technology, clocked at 800MHz. Table 5 shows that the 4/2-bit implementation achieves up to 2.56x lower silicon area and 5.56x lower power compared to that of the 8-bit baseline.

## 5 CONCLUSIONS

In this paper, we introduced a learned linear symmetric quantization (LLSQ) to quantize the whole network including the bias and scaling factors. We also use *BN* fusion and low bitwidth accumulators to reduce the network inference overhead and the hardware resources in integer neural accelerators. We show that our proposed method performs well for various networks on Cifar10, ImageNet, and Pascal VOC datasets. We also show that even the linear symmetric quantizer can obtain better results than asymmetric or non-linear quantization in the case of 4-bit networks. Finally, we deploy the quantized network onto our specialized integer-only neural network accelerator. Currently, the bitwidth of every layer in a network is all the same. Prior researches empirically find that different layers have different sensitivity to bitwidth of quantization. Hence in the future, we will explore a framework to support more flexible bitwidth for different layers or finer-grained quantization.

ACKNOWLEDGMENTS

This work was supported in part by the National Natural Science Foundation of China under Grant 61874124 and Grant 61902375.

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

## APPENDIX

### IMAGENET DETAILED

We conduct experiments on AlexNet, ResNet18, ResNer34, and MobileNetv2. For all of the experiments, we adopt channel-wise quantization for *Conv* layers and layer-wise quantization for *FC* layers as well as the activations. The AlexNet architecture is the same as the PyTorch Model Zoo,

Table 6: Train Time of ResNet18

| #Training Process | training time |
|---|---|
| Train the fp32 network from scratch | 1.0x |
| Quantize $w/a$ to 4/4 according to Step1 of Alg. 1 | 0.69x |
| Quantize $w/a$ to 3/3 according to Step1 of Alg. 1 | 0.69x |
| Quantize $b/\alpha$ to 8/8 according to Step2 of Alg. 1 | 0.01x |

and it consists of five *Conv* layers, three *FC* layers, three *MaxPool* layers, and two *Dropout* layers. To prevent over-fitting, we keep the *Dropout* layers when quantizing AlexNet. As shown in Figure 5d, we use the same learning rate scheduler for all experiments on ImageNet. The test curves are also shown in Figure 5. As we begin with the pre-trained full-precision weights, the test accuracy is already acceptable after one-epoch training. The final results are listed in Table 7.

**Training time.** The proposed LLSQ requires about 2/3 training epochs than that of floating-point network training. In each training iteration, LLSQ needs extra computation cost to optimize the quantizers. Specifically, the simulated gradients generation of the scaling factors is the major cost. Table 6 shows the total training time comparison of ResNet18 network. The quantization training time is 70% of baseline only.

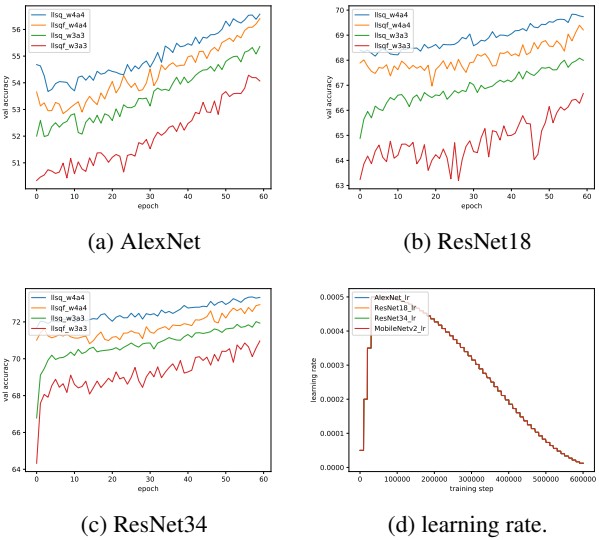

(a) AlexNet       (b) ResNet18

(c) ResNet34       (d) learning rate.

Figure 5: Test curves for AlexNet, ResNet18, and ResNet34 on ImageNet.

THE LLSQ ALGORITHM

Generate simulate gradients for $\alpha$:

$$E_m = \sum_i (\boldsymbol{x}_i^r - quantize_k(\boldsymbol{x}_i^r \mid \alpha))^2$$

$$E_l = \sum_i (\boldsymbol{x}_i^r - quantize_k(\boldsymbol{x}_i^r \mid \frac{\alpha}{2}))^2$$

$$E_r = \sum_i (\boldsymbol{x}_i^r - quantize_k(\boldsymbol{x}_i^r \mid 2\alpha))^2 \tag{9}$$

$$d_{better} = \arg\min([E_l, E_m, E_r]) - 1$$

$$\Delta G_\alpha = -\alpha^2 d_{better}$$

where $\boldsymbol{x}^r$ is one kernel of weights or one layer of activations. $\arg\min([E_l, E_m, E_r]) \in \{0, 1, 2\}$ selects the index of the smallest number in the array $[E_l, E_m, E_r]$.

The bit-shift quantization can be formulated as:

$$\boldsymbol{\alpha}^q = SQ_k(\boldsymbol{\alpha})$$
$$= \frac{\text{clamp}(\text{round}(2^{qcode} \cdot \boldsymbol{\alpha}), -2^{k-1}, 2^{k-1} - 1)}{2^{qcode}} \tag{10}$$

where $\boldsymbol{\alpha} \in \mathbb{R}^{+len(\boldsymbol{\alpha})}$ is the scaling factors to be quantized, $k \in \mathbb{Z}$ is the bitwidth, and $qcode \in \mathbb{Z}$ is the parameter of the bit-shift quantizer simply obtained by:

$$qcode = k - \text{ceil}(\log_2(\max(\boldsymbol{\alpha})) + 1 - 10^{-5}) \tag{11}$$

---

**Algorithm 1** LLSQ

---

**Input:** Dataset $(\boldsymbol{x}, \boldsymbol{y})$, where $\boldsymbol{x}$ is input and $\boldsymbol{y}$ is label; Pre-trained full-precision parameters $(\boldsymbol{w}, \boldsymbol{b})$, where $\boldsymbol{w}$ is weights and $\boldsymbol{b}$ is bias; Suppose the network consists of $L$ layers, $\boldsymbol{w}_l^{(i)}$ represents the $i_{th}$ kernel of weights of the $l_{th}$ layer while $\boldsymbol{a}_l$ is the output of the $l_{th}$ layer.

**Output:** The quantized scaling factors:
$$\hat{\boldsymbol{\alpha}}_{\boldsymbol{w}}^q = [[\hat{\alpha}_{\boldsymbol{w}_0^{(0)}}^q, \dots], \dots, [\hat{\alpha}_{\boldsymbol{w}_{L-1}^{(0)}}^q, \dots]]$$
$$\boldsymbol{\alpha}_{\boldsymbol{a}}^q = [\alpha_{\boldsymbol{a}_0}^q, \dots, \alpha_{\boldsymbol{a}_{L-1}}^q];$$
The quantized weights $\boldsymbol{w}^q$ and bias $\hat{\boldsymbol{b}}^q$.

***Step 1: Quantize weights and activations and Re-training***
// Re-training of quantized networks can converge faster and end with a higher accuracy due to the mechanism of *BN* layers.
**repeat**
 **Forward:**
 $\boldsymbol{a}_0 \leftarrow$ input
 **for** $l = 1, \cdots, L$ **do**
  **for** $\boldsymbol{w}_l^{(i)}$ in $\boldsymbol{w}_l$ **do** // This is accelerated in parallel when implemented.
   $\boldsymbol{w}_l^{(i)q} \leftarrow$ Quantize$(\boldsymbol{w}_l^{(i)} \mid \alpha_{\boldsymbol{w}_l^{(i)}})$ per Eq. (1)
   Generate simulated gradients for $\alpha_{\boldsymbol{w}_l^{(i)}}$ per Eq. (9)
  **end for**
  $\boldsymbol{w}_l^q \leftarrow \underset{i}{\text{Concat}} \; \boldsymbol{w}_l^{(i)q}$
  $\boldsymbol{a}_l \leftarrow ReLU(BN(Conv(\boldsymbol{a}_{l-1}^q, \boldsymbol{w}_l^q, \boldsymbol{b}_l)))$
  $\boldsymbol{a}_l^q \leftarrow$ Quantize$(\boldsymbol{a}_l \mid \alpha_{\boldsymbol{a}_l})$ per Eq. (1)
  Generate simulated gradients for $\alpha_{\boldsymbol{a}_l}$ per Eq. (9)
 **end for**
 **Backward:**
 Generate $\Delta G$ for weights and bias and $\Delta E$ for activation per Eq. (7), (8) and backpropagation algorithm.
 Update $\boldsymbol{w}, \boldsymbol{b}, \boldsymbol{\alpha}_{\boldsymbol{w}}, \boldsymbol{\alpha}_{\boldsymbol{a}}$
 $iter \leftarrow iter + 1$
**until** $iter \geq iter_{max}$ // need about 60 epochs, e.g. $iter_{max} = 60\frac{\text{len}(dataset)}{batchsize}$

***Step 2: Quantize bias and scaling factor after BN fusion and Re-training***
$iter \leftarrow 0$
$\hat{\boldsymbol{\alpha}_w}, \hat{\boldsymbol{b}} \leftarrow$ BN fusion per Eq. (5)
**repeat**
 **Forward:**
 $\boldsymbol{a}_0 \leftarrow$ input
 **for** $l = 1, \cdots, L$ **do**
  $\hat{\boldsymbol{\alpha}}_{\boldsymbol{w}_l}^q \leftarrow$ SQ$(\hat{\boldsymbol{\alpha}}_{\boldsymbol{w}_l})$ per Eq. (10)
  **for** $\boldsymbol{w}_l^{(i)}$ in $\boldsymbol{w}_l$ **do** // This is accelerated in parallel when implemented.
   $\boldsymbol{w}_l^{(i)q} \leftarrow$ Quantize$(\boldsymbol{w}_l^{(i)} \mid \hat{\alpha}_{\boldsymbol{w}_l^{(i)}}^q)$ per Eq. (1)

$\hat{b}_l^{(i)q} \leftarrow$ Quantize($\hat{b}_l^{(i)} \mid \alpha_{\boldsymbol{a}_{l-1}}^q \hat{\alpha}_{\boldsymbol{w}_l^{(i)}}^q$) per Eq. (1)

Generate simulated gradients for $\hat{\alpha}_{\boldsymbol{w}_l^{(i)}}$ per Eq. (9)

**end for**

$\boldsymbol{w}_l^q \leftarrow \underset{i}{\text{Concat }} \boldsymbol{w}_l^{(i)q}$

$\hat{\boldsymbol{b}}_l^q \leftarrow \underset{i}{\text{Concat }} \hat{b}_l^{(i)q}$

$\boldsymbol{a}_l \leftarrow ReLU(Conv(\boldsymbol{a}_{l-1}^q, \boldsymbol{w}_l^q, \hat{\boldsymbol{b}}_l^q))$

$\alpha_{\boldsymbol{a}_l}^q \leftarrow$ SQ($\alpha_{\boldsymbol{a}_l}$) per Eq. (10)

$\boldsymbol{a}_l^q \leftarrow$ Quantize($\boldsymbol{a}_i \mid \alpha_{\boldsymbol{a}_l}^q$) per Eq. (1)

Generate simulated gradients for $\alpha_{\boldsymbol{a}_l}$ per Eq. (9)

**end for**

**Backward:**

Generate $\Delta G$ for weights and bias and $\Delta E$ for activation per Eq. (7), (8) and backpropagation algorithm.

Update $\boldsymbol{w}, \hat{\boldsymbol{b}}, \hat{\boldsymbol{\alpha}}_w, \boldsymbol{\alpha}_a$

$iter \leftarrow iter + 1$

**until** $iter \geq iter_{max}$ // only need one epoch, e.g. $iter_{max} = \frac{\text{len}(dataset)}{batchsize}$

Table 7: Comparison with state-of-the-art quantization methods on ImageNet. Top1, Top5 accuracy(%) and degradation of Top1 are given.

| Method | Model | # Bits $w/a/b/\alpha$ | Top1(%) | Top5(%) | Degradation(%) |
|---|---|---|---|---|---|
| LQ-Nets[*] | ResNet18 | Reference | 70.3 | 89.5 | |
| (Zhang et al., 2018a) | | 4/4 | 69.3 | 88.8 | 1.0 |
| | | 3/3 | 68.2 | 87.9 | 2.1 |
| RQ | ResNet18 | Reference | 69.54 | 89.19 | |
| (Louizos et al., 2019) | | 8/8 | 69.97 | 89.44 | -0.43 |
| | | 4/4 | 61.52 | 83.99 | 8.02 |
| RQ+ST | | 8/8 | 69.63 | 89.33 | -0.09 |
| (Louizos et al., 2019) | | 4/4 | 62.46 | 84.78 | 7.08 |
| LLSQ(ours)[*] | ResNet18 | Reference | 69.76 | 89.08 | |
| | | 4/4 | 69.84 | 89.14 | **-0.08** |
| | | 3/3 | 68.08 | 88.20 | **1.68** |
| LLSQF(ours) | ResNet18 | 4/4 | 69.40 | 88.72 | **0.36** |
| | | 3/3 | 66.67 | 87.42 | 3.09 |
| LQ-Nets[*] | ResNet34 | Reference | 73.80 | 91.40 | |
| (Zhang et al., 2018a) | | 3/3 | 71.9 | 90.2 | 1.9 |
| LLSQ(ours)[*] | ResNet34 | Reference | 73.30 | 91.42 | |
| | | 4/4 | 73.60 | 91.28 | -0.30 |
| | | 3/3 | 72.02 | 90.66 | **1.28** |
| LLSQF(ours) | ResNet34 | 4/4 | 72.94 | 91.20 | 0.36 |
| | | 3/3 | 70.97 | 89.95 | 2.33 |
| LLSQ(ours)[*] | AlexNet | Reference | 56.55 | 79.09 | |
| | | 4/4 | 56.57 | 79.02 | -0.02 |
| | | 4/4/8/8 | 56.45 | 80.15 | 0.10 |
| | | 3/3 | 55.36 | 78.20 | 0.19 |
| LLSQF(ours) | AlexNet | 4/4 | 56.40 | 78.85 | 0.15 |
| | | 4/4/8/8 | 55.58 | 77.47 | 0.97 |
| | | 3/3 | 54.28 | 77.65 | 2.27 |
| RQ | Mobilenet(v1) | Reference | 70.61 | 89.47 | |
| (Louizos et al., 2019) | | 8/8 | 70.43 | 89.42 | 0.18 |
| | | 6/6 | 68.02 | 88.00 | 2.59 |
| | | 5/5 | 61.38 | 83.73 | 9.23 |
| RQ+ST | Mobilenet(v1) | 8/8 | 70.06 | 89.52 | 0.55 |
| (Louizos et al., 2019) | | 6/6 | 67.62 | 87.78 | 2.99 |
| | | 5/5 | 56.85 | 80.35 | 13.76 |
| LLSQ(ours)[*] | MobileNet(v2) | Reference | 71.80 | 90.37 | |
| | | 6/6 | 71.20 | 89.99 | 0.60 |
| | | 5/5 | 70.45 | 89.69 | 1.35 |
| | | 4/4 | 67.37 | 87.99 | 4.43 |

[*] First and last layers in full precision.

