# OpenReview forum: "Linear Symmetric Quantization of Neural Networks for Low-precision Integer Hardware"
_ICLR.cc/2020/Conference — Accept (Poster)_

### Official Review · AnonReviewer3 · 2019-10-23
**Official Blind Review #3**

**Rating:** 6

**Review:**

The submission proposes to train a linear symmetric quantizing function for integer processors.

The proposed idea makes sense at a high level, and the empirical results look somewhat compelling, but the write-up is not particularly clear (see clarity-related comments). I found it hard to extract a complete picture of how the proposed approach operates: from the leftmost diagram in Figure 1 I can infer what high-level steps are involved, but I wouldn’t know how to re-implement the approach from the textual description itself.

Clarity-related comments:

- The submission never explicitly states what x^r and p(x^r) correspond to in the network. My understanding from the context is that x^r is the scalar value of a model parameter or activation, and that p(x^r) is the empirical distribution over all model parameters and activations. Can the authors clarify?
- Some kind of pseudo-code for the re-training of the weights and activations (Section 3.3) would help clear up reader confusion. At this point, are only the weights and activations quantized in the network? How is backpropagation handled (and if the straight-through estimator is used, why is it only mentioned in Section 3.5)? How do parameter and alpha updates interleave?
- How many alpha values are there? Section 3.3 gives the impression that there is a single quantization parameter shared by all parameters and activations, but Equation 3 uses different alpha values for the weights and activations.
- Section 3.4 is difficult to parse due to bad notation. If the output of the convolution layer is fed into the batch normalization layer, it would be clearer to reuse symbols (i.e. change “x” to “o” in Equation 4).

Additional comments:

- Organization of the different sections could be improved. Related work discussion is scattered throughout three different sections, namely Introduction, Motivation, and Related Work.
- Writing quality could be improved  (e.g., “*Except that*, some designs rely [...]”, “Edge or embedded neural network accelerators *are generally having* three primary design goals [...]”) but has a relatively small negative impact on readability.

**Experience Assessment:**

I have read many papers in this area.

**Review Assessment: Checking Correctness Of Derivations And Theory:**

N/A

**Review Assessment: Checking Correctness Of Experiments:**

I assessed the sensibility of the experiments.

**Review Assessment: Thoroughness In Paper Reading:**

I read the paper at least twice and used my best judgement in assessing the paper.

---

> ### Author Response · Authors · 2019-11-11
> **Pseudo-code and Code are available**
>
> We thank the reviewer for their time in looking at our paper, and we appreciate their feedback.
>
> “I wouldn’t know how to re-implement the approach from the textual description itself”
> R: The pseudo-code for the re-training of the weights, activations, bias, and scaling factors is summarized in the paper. And the training code for LLSQ is also available at
> https://anonymous.4open.science/r/c05a5b6a-1d0c-4201-926f-e7b52034f7a5/
> The parameters and alpha values update strategy is also summarized in pseudo-code.
>
> “$x^r$ and $p(x^r)$”
> R: $x^r$ is indeed the model parameters or activations to be quantized, and $p(x^r)$ is the probability density distributions of weights or activations. Thanks for the reviewer points it out, and we have updated it in the submission.
>
> “are only the weights and activations quantized in the network”
> R: The experimental results are summarized in Table 3 and Table 7 in the paper. And the bit-width of weights (w), activations (a), bias(b), and scaling factor (alpha) are given. “4/4” donates that we only quantize the weights and activations into 4-bit integers while “4/4/8/8” indicates that we not only quantize the weights and activations into 4-bits but also quantize bias and scaling factor into 8-bit.
>
> “How many alpha values are there?”
> R: We use channel-wise quantization for convolutional layers and layer-wise quantization for fully-connected layers and activations. For example, suppose there are 32 output-channels for one convolutional layer, there are 32 kernels(filters) of weights and 32 alpha values for each kernel. And For activations, each layer has only one alpha value. Please note that the alpha values are not shared across layers because the weights and activations distributions of different layers are different. The equations in Section 3.3 correspond to a weight kernel or a layer of activations. Due to the quantizer optimization algorithm is the same for each kernel of weights or each layer of activations, we present a general formalization in the paper only.  However, Section 3.3 is indeed somewhat misleading. We have revised the corresponding content in Section 3.3.
>
> “Section 3.4 is difficult to parse due to bad notation.” And “Additional comments”
> R: Thanks for the reviewer’s suggestion. We modified the paper so that the reader can better understand.

---

> > ### Comment · AnonReviewer3 · 2019-11-13
> > **Follow-up**
> >
> > Thank you for adding a pseudocode algorithm in the Appendix and providing a link to the source code. The pseudocode clears up some of my confusion and improves the submission's clarity. I will update my score accordingly.
> >
> > > We use channel-wise quantization for convolutional layers and layer-wise quantization for fully-connected layers and activations. For example, suppose there are 32 output-channels for one convolutional layer, there are 32 kernels(filters) of weights and 32 alpha values for each kernel. And For activations, each layer has only one alpha value.
> >
> > Thank you for clarifying. From your response I gather that channel-wise and layer-wise quantization necessarily implies separate quantization parameters for individual channels and layers; this is perhaps obvious in hindsight, but maybe it would be worth stating this fact explicitly to clear up any possible confusion?

---

### Official Review · AnonReviewer2 · 2019-10-24
**Official Blind Review #2**

**Rating:** 6

**Review:**

This paper proposes a linear symmetric quantizer for integer accelerators called LLSQ, which learns the quantization scaling factor using simulated gradient as update policy. Their main contribution is enabling inference on integer-only hardware by covering all parameters of all operators in convolutional networks, including weight, bias, activation and scaling factor. To address the quantization noise issue in bias parameters, they adopt Straight-Through Estimator and fine-tune the parameters after quantization. To improve inference efficiency, they apply BN layer fusion. They conduct experiments on public datasets for image classification and object detection to conclude that LLSQ achieves lower accuracy degradation compared to previous work. Finally, they test the quantized model on a specialized integer accelerator, showing the feasibility of the quantization on real hardware.

In conclusion, this paper generalizes linear symmetric quantization to all parameters in order to deploy the network on specialized integer neural network processors for efficient inference. In terms of algorithmic contribution, this paper introduces the scaling factor as a learnable parameter of the quantizer, but lacks enough theoretical justification. Therefore, I would consider weakly accepting the paper.

For the algorithm, the following should be addressed.
1.	In scaling factor updating policy, the simulated gradient performs better compared to EMA. However, the motivation for choosing this policy is unclear, and there is no mathematical derivation of the gradient value.
2.	In this study, pretrained network weights in full precision are used, thus the quantization procedure should take fixed weights as inputs. However, in section 3.3, the quantization scaling factor is updated in the training phase of the network, with weights being updated at the same time This seems to be a contradiction, and the experiment results where quantizer is trained with network weights fixed should also be presented.

For the experiment, the following should be addressed.
1.	The training time for different bit-width settings should be included in the results.
2.	The reason for leaving the first and last layers in full precision is unclear, and doing so may go against the objective of deploying the quantized model on specialized integer hardware according to the paper.


**Experience Assessment:**

I do not know much about this area.

**Review Assessment: Checking Correctness Of Derivations And Theory:**

I assessed the sensibility of the derivations and theory.

**Review Assessment: Checking Correctness Of Experiments:**

I did not assess the experiments.

**Review Assessment: Thoroughness In Paper Reading:**

I read the paper at least twice and used my best judgement in assessing the paper.

---

> ### Author Response · Authors · 2019-11-11
> **Reply to Official Blind Review #2**
>
> We thank the reviewer for their time in looking at our paper, and we appreciate their feedback.
>
> “1. The motivation for choosing simulated gradient method”
> R: As for the scaling factors, we hope to update them fast in the early stages of training and gradually stabilize them in the later stages of training to achieve better performance (e.g., “accuracy” for classification tasks). The EMA method takes advantage of the old values but does not change the update speed during training. However, we can use the learning rate to control the updating speed of scaling factors if the simulated gradients are given. This is the motivation for choosing the SG method.
>
> “2. The scaling factors and weights are updated at the same time”
> R: Although we use pre-trained full-precision parameters, we re-train the parameters and the scaling factors in the training phase. This paper focuses on quantization-aware training [1,2,3] while other works [4] that only train the quantizers are post-training quantization. In general, the accuracy degradation of 4-bit post-training quantization is significant.
>
> “The training time for different bit-width settings”
> R: The results of the training time will be included in the paper as soon as possible.
> R2: The results of the training time have been updated in the paper.
>
> “The reason for leaving the first and last layers in full precision”
> R: Leaving the first and last layers in full precision is a common feature of prior work on network quantization, which cannot be run on popular integer machine processors and is also why we need a new quantizer like LLSQ in this work to make it work. For a fair comparison with previous state-of-the-art quantization methods [2], which leave the first and last layers in full-precision, we adopt the same settings and show the results of LLSQ with full-precision first and last layers. However, we quantize all of the parameters and scaling factors in the networks when deploying models to the specialized integer hardware. Showing results of baselines and LLSQ with full-precision first and last layers, as well as that of the hardware-friendly LLSQ with all quantized layers, is believed to help the readers to better understand the effects of the proposed solution.
>
> [1] Benoit Jacob, et al. Quantization and Training of Neural Networks for Efﬁcient Integer-Arithmetic-Only Inference. CVPR2018
> [2] Dongqing Zhang, et al. LQ-Nets: Learned Quantization for Highly Accurate and Compact Deep Neural Networks. ECCV2018
> [3] Christos Louizos, et al. Relaxed quantization for Discretized Neural Networks. ICLR2019
> [4] Markus Nagel, et al. Data-Free Quantization through Weight Equalization and Bias Correction.

---

### Official Review · AnonReviewer1 · 2019-11-05
**Official Blind Review #1**

**Rating:** 3

**Review:**

This paper focuses on the quantization of ConvNets. This paper proposes a learned linear symmetric quantizer to reduce the precision of weight, bias, and activation. The proposed approach works as the following: for a pre-trained neural network, it computes the new weight and activation as a product of a quantized value with a scaling factor. The quantization is based on a simple linear, symmetric function as in equation (1). The value of the scaling factor is searched by "simulated gradient" or exponential moving average during re-training. Next, batch normalization is fused into convolution, and the scaling factor and biases are re-calculated. Last, the scaling factor on the convolution is merged with bias terms, to remove the need for multiplication in hardware implementation. Since bias terms usually have a much larger dynamic range, higher precision is used to represent biases. Experiments show that the method achieves competitive results compared with previous quantization methods, and the quantized models can be deployed on hardware more easily.

The contribution of the paper, in my opinion, is to show that using simple methods without many bells and whistles, we can achieve competitive quantization performance.  And when performing quantization, it is important to consider the hardware implementation. Details on how to deal with scaling factors, how to deal with biases, and so on, can have significant influences on the overall performance.

However, the main concern of the paper is that the methods adopted in the paper are too plain. The paper successfully integrated previous methods but did not propose new ideas that inspire future research. As a result, I would not recommend acceptance for publication.

**Experience Assessment:**

I have published one or two papers in this area.

**Review Assessment: Checking Correctness Of Derivations And Theory:**

I assessed the sensibility of the derivations and theory.

**Review Assessment: Checking Correctness Of Experiments:**

I assessed the sensibility of the experiments.

**Review Assessment: Thoroughness In Paper Reading:**

I read the paper at least twice and used my best judgement in assessing the paper.

---

> ### Author Response · Authors · 2019-11-11
> **Reply to Official Blind Review #1**
>
> We thank the reviewer for their patience in providing value feedbacks.
>
> In contrast to previous quantization work, this work addresses a not-solved-yet problem with a simple but effective method. The motivation of this work is to deploy the quantized models to the integer accelerators, which are popular low-power hardware solutions to neural networks in the market. We not only achieve BETTER results than previous quantization work, and also succeed to eliminate the complicated operators like first and last layer treatment and the scaling layers, which cannot be handled by off-the-shelf integer accelerators. Thus, using simple methods and operators is not just to show another viable technique of quantization, but is MANDATARY for smooth running on the integer accelerators, which cannot be satisfied by previous quantizers. During network quantization, there are many other constraints in such hardware to meet, which is the essential motivation of this work. For example, Non-linear quantization needs LUT operations rather than low-precision multipliers, while asymmetric quantization requires additional subtraction [1] before multiplication. As a result, we adopt the symmetric linear quantization. To our best knowledge, eliminating the Batch-Norm layers can bring performance improvements to accelerators, while many other low bit-width (4-bit or lower) quantization methods didn’t consider or solve this problem. In contrast, LLSQ can easily adopt BN fusion and achieve state-of-the-art accuracy. In addition, the remanding values in the neural networks are also quantized to low bit-width integers. Finally, we are able to deploy the quantized models to the low-bit integer accelerators.
>
> [1] Benoit Jacob, et al. Quantization and Training of Neural Networks for Efﬁcient Integer-Arithmetic-Only Inference. CVPR2018

---

### Author Response · Authors · 2019-11-11
**New supplemental materials**

New supplemental materials have been added to the appendix of the paper, and we will continue to update the paper before the rebuttal deadline.

Mainly update:
1. The pseudo-code;
2. The training time of LLSQ;
3. The simulated gradient formulation of the scaling factors and the bit-shift quantization formulation.

---

### Decision · Program_Chairs · 2019-12-19

**Decision:**

Accept (Poster)

**Comment:**

This paper considers the question of how to quantize deep neural networks, for processors operating on low-precision integers.  The authors propose a methodology and have evaluated it thoroughly. The reviewers all agree that this question is important in practice, though there was disagreement about how novel a contribution this paper is specifically, and on its clarity. The clarity questions were resolved on rebuttal, so I lean to accepting the paper.